# Culinary Comfort: Understanding the Connection between Food and Well-Being

**DOI:** 10.3390/nu16121865

**Published:** 2024-06-14

**Authors:** Bushra Yasmeen, Ifra Iftikhar, Florian Fischer

**Affiliations:** 1School of Sociology, Minhaj University, Lahore 54770, Pakistan; bushrayasmin.soc@mul.edu.pk; 2Department of Mass Communication, Lahore Garrison University, Lahore 54000, Pakistan; ifraiftikhar@lgu.edu.pk; 3Institute of Public Health, Charité—Universitätsmedizin Berlin, 10117 Berlin, Germany

**Keywords:** Culinary Comfort and Connection Index (CCCI), social–ecological model, well-being, relationships dynamics

## Abstract

This study investigated the complex interaction between individuals’ culinary tastes, at-home dining habits, and their broader impact on general well-being and relationships dynamics. An all-encompassing tool for assessing the impact of dietary choices on emotions related to coziness, social interaction, and general wellness, the multidimensional Culinary Comfort and Connection Index (CCCI) focuses on traditional home-cooked meals, in particular. We used an online-based survey to validate the CCCI. With a Cronbach alpha of 0.844, this scale is reliable and valid. It covers a wide range of aspects including self-care advocacy, traditional values, and a preference for handmade food. We performed descriptive and stratified analyses and tested correlations. The CCCI shows complicated patterns when analyzed with respect to gender, education level, and family income that demonstrate a myriad of factors impacting people’s views on food and its relationship to health. While some patterns emerged, the results imply that dietary choices do not necessarily correlate with overall health. The research highlights the complex interaction between cultural, societal, and personal elements in determining perspectives on nutrition and health by drawing on theoretical models like Bronfenbrenner’s ecological systems theory and the Theory of Planned Behavior. Future research should incorporate broader age ranges, longitudinal designs, different populations, objective measurements, and intervention trials to better understand the dynamic link between dietary preferences and health outcomes.

## 1. Introduction

There has been a change in eating habits over the last several decades, with less time spent on shopping for food, preparing meals at home, and eating out, even though these activities may have health benefits [1]. The rise in diet-related diseases like diabetes and obesity has been accompanied by a decline in the amount of time people spend in the kitchen and changes in eating habits [2].

Health professionals and public health experts are looking for practical solutions to tackle the obesity epidemic. One viable strategy is for more people to eat at home and share meals with their families. According to certain studies, home-cooked meals may positively impact children’s health, mental and social development, and family ties [3]. People of all ages, nationalities, and socioeconomic backgrounds benefit greatly from eating regular meals together as a family [4]. For instance, evidence suggests that having family meals together can help adolescents with their self-esteem, their academic performance, and substance misuse prevention [5]. However, there needs to be a current evidence review on the positive effects of home-cooked meals on family connections, health outcomes, social variables, or the quality of one’s diet or meal patterns.

Previous research has shed light on four primary areas related to eating at home: (1) lifestyle factors and family demographics that may influence eating at home [6]; (2) how eating at home affects the nutritional quality of meals [7]; (3) how eating at home affects the health outcomes of young adults [8]; and (4) how eating at home affects family relationships [9,10]. However, no single index has been created to check the family culinary habits and their association with all these areas. This study addresses the gap related to the overall comfort and connection that young adults feel regarding family meals together, including among university-going young adults.

In children, adolescents, and young adults, studies conducted in the past decade have demonstrated that eating meals together as a family protects against unhealthy weight-related outcomes (such as excessive weight gain and disordered eating behaviors) [10,11], substance use [12], and poor psychosocial outcomes [5,13]. As it provides a daily chance for healthy eating and connection between family members, dining together is believed to provide physical and psychosocial benefits for people of all ages [14]. Several prestigious groups, associations, and researchers have voiced their support for frequent family meal participation as a means to shield children from a variety of negative health outcomes, based on the available evidence [15].

Evidence indicates that regular family mealtimes positively affect health and happiness, as well as the quality of relationships within the family [16]. Proposed mechanisms include consistent positive parenting, few instances of open conflict, and well-regulated emotions resulting from recurrent social encounters. It shows a correlation between family mealtime frequency and overall family functioning [17]. According to a longitudinal study [18], eating together as a family may help teenagers improve their communication skills. Throughout three and a half years, the researchers discovered that the frequency with which parents and children communicated was correlated with how often they ate family dinners when they were younger. There was less deterioration in parent–child contact among teenagers from homes that began with more frequent mealtimes [18]. Mealtime frequency and its predicted impacts on family dynamics have only been studied in a few longitudinal studies.

The primary objective of this study was to examine the correlation between various groups in the Culinary Comfort and Connection Index (CCCI) and demographic factors such as gender, education level, and family income. Additionally, the study aimed to investigate the relationship between individuals’ attitudes toward nutrition, their personal health status, and the aforementioned demographic factors based on the categorizations within the CCCI groups. The specific aim of this investigation was to analyze the impact of gender, education level, and family income on the distribution of individuals in low, medium, or high CCCI groups.

### 1.1. Hypotheses of the Study

This study proposed three primary hypotheses to investigate the correlations between demographic parameters and categorization into different Culinary Comfort and Connection Index (CCCI) groups, looking at gender, education, and family income as potential influencing factors.

*Gender hypothesis:* The study expected that gender significantly affects categorizing people into low, medium, or high CCCI groups. It expected women to be more connected to their diets and well-being than men.

*Education hypothesis:* Undergraduate and postgraduate education levels were expected to correlate significantly with CCCI group classification. Higher-educated people were predicted to have a stronger correlation between eating and well-being.

*Family income hypothesis:* The study hypothesized that family income affects participants’ distribution among low, medium, and high CCCI categories. It predicted that higher-income households would have a weaker link between diet and health than lower-income families.

### 1.2. Theoretical Framework

The complex interplay between people and their natural surroundings can be better understood with the help of Bronfenbrenner’s ecological systems theory [19]. Several aspects of our research on the CCCI and its correlation with well-being can be mapped onto Bronfenbrenner’s theory.

According to Bronfenbrenner’s ecological systems theory [19], the development of an individual is impacted by several linked environmental systems. These systems range from the local surroundings (such as the family) to broad social structures (such as culture). These systems consist of the microsystem, the mesosystem, the ecosystem, the macrosystem, and the chronosystem. Each of these systems represents a distinct level of environmental influence on the development and behavior of an individual [20].

The microsystem, comprised of the immediate and direct settings in which an individual lives, plays a significant part in forming perceptions and behaviors. Our research shows that people’s dwellings and food choices make up the microsystem surrounding them. The microsystem’s influence is immediately reflected in the CCCI, which evaluates feelings around food and the impact it has on well-being.

Mesosystems are made up of the interconnections that exist between the components of microsystems. How many aspects of the local environment, such as home-cooked meals, family dynamics, and social bonds, all interact with one another was how our research established this point. We can observe the influence of the mesosystem in the manner in which these interconnected components contribute to the overall sense of enjoyment that people experience.

The ecosystem is comprised of external factors that have an indirect impact on the members of the group. In the context of our investigation, the external setting may include broader societal implications on dietary choices and overall health. One factor that may indirectly influence the connection between culinary tastes and general well-being is the influence of cultural norms, society expectations, and economic circumstances.

The macrosystem represents the larger social and cultural setting. The macrosystem may entail more significant cultural perspectives on food, health, and overall well-being for our investigation. Individuals’ judgments of the relevance of their food choices to their overall health and happiness may be shaped by other factors, such as cultural influences, traditions, and societal ideals.

In the chronosystem, the influence of time on development is taken into consideration. Inside the scope of our investigation, the age range of 18 to 30 years is considered to be inside the chronosystem. Throughout their lives, people in this age group may encounter a variety of life phases, experiences, and cultural shifts, all of which may have an impact on how they view the connection between their food choices and their overall health.

Bronfenbrenner’s ecological systems theory fully grasps how the microsystem, mesosystem, ecosystem, macrosystem, and chronosystem interact to influence people’s attitude toward food and its effects on their health. This research has a solid theoretical foundation that supports the idea that studying human development requires considering the complex interplay between people and their surroundings.

## 2. Methods

An online-based survey was used for this study using a convenience sampling strategy targeting persons aged 18–30 years. We developed a questionnaire for this study (see Appendix A). In the study, participants’ age, capacity to take an online survey, permission, language competence, and cognitive ability to understand and answer survey questions were assessed. Exclusion criteria included people that were outside the age range, were unable to finish the survey, could not provide informed consent, could not speak the survey language, and had cognitive disabilities that hindered correct responses. These criteria ensured that the sample met the study goals and provided accurate data for analysis. The study emphasized survey response accuracy and comprehension.

Participants were divided into groups based on demographic factors such as gender, education level, and household income. The CCCI was used as the main instrument, measuring individuals’ attitudes about food in relation to their general well-being and its relationship to demographic characteristics.

### 2.1. Culinary Comfort and Connection Index

A delicate interplay between people’s culinary preferences, at-home dining habits, and the wider impact on their general well-being and relational dynamics is captured by the multidimensional CCCI. It evaluates the degree to which dietary decisions—especially those associated with traditional, home-cooked meals [21]—support coziness, social interaction, self-esteem [22], and overall wellness [23]. Items included in the scale are as follows:Prefer homemade food: Denotes a preference for eating meals cooked at home as opposed to relying on outside or processed options;Home food flavors: Expresses a fondness for the unique flavors and comfort that come with cooking meals at home;Prefer traditional foods: Indicates a person’s inclination toward traditional and culturally grounded cuisine;Creates ties (strengthens relationships by creating a sense of trust, belongingness, and closeness);Enhances quality of life (provides energy, saves money): Assesses the wider lifestyle advantages of cooking at home, such as financial savings and increased energy;Creates an environment for conversation: Examines how eating can work as a social catalyst by fostering the kind of atmosphere that fosters deep dialogue and close family relationships;Promotes self-care (raises self-esteem): Looks at how a person’s dietary decisions can support their self-care routines and boost their self-esteem;Facilitates family care: Emphasizes a group approach to health and happiness and assesses how much dietary decisions assist family members’ care and well-being;Promotes a healthy way of living: Evaluates how dietary choices and lifestyle factors, including nutrition, affect general health and well-being;Increases focus and lessens stress: Highlights the positive effects of food choices on mental health and concentration while also examining the cognitive and stress-relieving aspects of eating.

### 2.2. Data Analysis

We performed descriptive and stratified analyses and tested correlations. A significance level of 0.05 was chosen. We used the Pearson chi-square test and the likelihood-ratio test.

The reliability of the scale was determined by the Cronbach alpha value of 0.844 (M = 1.77, SD = 0.72). Greater alignment between dietary choices and holistic well-being is indicated by a higher CCCI score, which highlights the role that traditional, home-cooked meals play in promoting coziness, family time, and general life satisfaction. This index offers a thorough understanding of the complex connections between dietary decisions and social, familial, and individual well-being.

## 3. Results

This study examined the associations between demographic characteristics (gender, education, and family income) and Culinary Comfort and Connection Index (CCCI) categories using chi-square testing.

### 3.1. Gender and CCCI

The relationship was examined between people’s feelings about food, as measured by the CCCI, and their general well-being, taking gender differences into account. Based on their CCCI ratings, participants were divided into three groups: low, medium, and high (Table 1). Notably, the low group, which accounts for approximately 15% of our study sample, includes those who do not believe there is a significant link between their dietary choices and overall well-being. A greater proportion, approximately 19%, appears to have a moderate link, indicating a proclivity to associate comfort or pleasurable experiences with the food they consume. A smaller subset, about 3.6%, closely links their dietary choices with their overall well-being, stressing the importance of food in their emotional state.

Table 1 shows that females are more prevalent in the medium and high comfort groups, implying that, on average, the females in our study perceive a stronger link between their culinary choices and general well-being than males.

To validate the disparity between both genders concerning CCCI classifications, the chi-square test was conducted. No significant connection was found between gender and CCCI categories, according to the findings of the Pearson chi-square test and the likelihood-ratio test (χ^2^(2) = 3.001, *p* = 0.223; χ^2^(2) = 3.194, *p* = 0.203, respectively). In addition, there was no apparent linear relationship between gender and the CCCI categories, as shown by the linear-by-linear association test (χ^2^(1) = 0.921, *p* = 0.337). We looked for predicted numbers more significant than 5 and less than 7.19 to satisfy the chi-square test assumptions. The results indicate that the CCCI was not significantly impacted by gender in our sample.

### 3.2. Education and CCCI

Taking into account variations in educational attainment, our study examined the relationship between people’s well-being and their feelings toward food as measured by the CCCI. Undergraduates and postgraduates were the two groups into which the participants were divided (Table 2). About 39.6% of our sample fell into the low CCCI category, with 27% being undergraduates and 12.6% being postgraduates. This group represents those who might not directly link their diet to their health.

For the medium CCCI group, which made up 43.2% of the study sample, 30.6% were undergraduates and 12.6% were postgraduates. This indicates that many people, especially college students, think there is a moderate link between what they eat and how they feel physically.

Overall, 12% were undergraduates and 5% were postgraduates among the participants falling into the high CCCI group. There was a considerable correlation between the dietary choices of this smaller minority and their overall well-being, suggesting that food significantly affects their emotional state.

Differences in CCCI classifications based on the level of education are further shown in Table 2. It is worth mentioning that out of the overall sample, 70.3% were undergraduates and 29.7% were postgraduates.

The chi-square test was used to confirm that there were significant differences in the CCCI categories according to degree of education. No significant correlation was found between education level and the CCCI categories. According to the results obtained from the Pearson chi-square test and the likelihood-ratio test (χ^2^(2) = 0.205, *p* = 0.903; χ^2^(2) = 0.207, *p* = 0.902, respectively). The lack of a noticeable linear association was further validated by the linear-by-linear association test (χ^2^(1) = 0.203, *p* = 0.652). The chi-square test was satisfied because all of the anticipated values were larger than 5, and none were less than 5.65. Taken together, these results indicate that study sample’s CCCI was unaffected by the participants’ degree of education.

### 3.3. Family Financial Status and CCCI

In the investigation of the correlation between people’s food-related emotions (as measured by the CCCI) and their general health, the study focus was shifted to variations in their families’ financial status. To understand the possible trends in our sample, Table 3 presents a cross-tabulation that breaks down the CCCI groups according to household income levels.

The complex relationship between people’s feelings about food and their financial situations can be better understood by distributing the CCCI categories across different family income levels. Remarkably, 39.6% of participants fell into the low CCCI category. This group appears to be evenly distributed across low, middle, and high family income, reflecting that people from different income levels believe a slight correlation exists between what they eat and how they feel physically. The medium CCCI group includes 43.2% of the sample and shows an apparent disparity; however, as it contains a disproportionately large number of people with low family incomes. Our data reveal that people from lower-income families are more prone to believe in a moderate connection between what they eat and their health. Across all income categories, the proportion of participants in the high CCCI category, representing 17.1% of the total, suggests that fewer people in higher-income groups strongly link their dietary choices to their overall health. The study cohort primarily consists of individuals with lower family incomes, as indicated by the bulk of the sample falling into low family income, followed by middle family income, and, finally, high family income.

The observed distribution of CCCI categories across family income levels is likely due to chance, given that the p-values associated with the Pearson chi-square, likelihood-ratio, and linear-by-linear association tests are all greater than 0.05. A minimum expected count of 4.45 was assumed, while one cell (11.1%) had a scheduled count below 5. This is acceptable considering the total sample size, even though it is below the traditional criterion. The results of the chi-square tests indicate that there appears to be no statistically significant correlation between household income and CCCI classifications.

## 4. Discussion

Intriguing patterns emerged from the study sample when examining the association between gender and the CCCI categories. No statistically significant connection was found in the chi-square tests, even though there were differences in the distribution of CCCI categories between the sexes. There appears to be a more significant perceived connection between dietary choices and overall health, as more females falls into the medium and high CCCI groups. On the other hand, for the specified age range (18–30 years), gender does not appear to have a statistically significant effect on the CCCI.

Concerning CCCI and education, the research divided its subjects into undergraduates and postgraduates. Although there were differences in the distribution of CCCI categories between these groups, no significant link was found using chi-square testing. The results show that educational level within the given age range had no effect on the correlation between feelings about food and general happiness.

The inquiry into family income levels highlights trends in the distribution of CCCI categories. The fact that people of all income brackets fall into the low CCCI category shows a small but statistically significant relationship between food and health. On the other hand, there is an overrepresentation of low-income families in the medium CCCI category, suggesting that this demographic is more likely to believe in the link between diet and health. Fewer people in the higher-income bracket closely link their food choices to general health as the high CCCI category declined across all income brackets. Regardless of these tendencies, there is no statistically significant relationship between the CCCI categories and family income, according to the chi-square tests. Previous research has shown no statistically significant correlation between sociodemographic characteristics and the frequency of family mealtimes; our results are in line with that [3,10].

By highlighting the interdependence of personal, interpersonal, communal, and societal elements, Bronfenbrenner’s ecological systems theory highlights the complexity of human behavior. This concept proposes that cultural norms, social networks, and individual characteristics contribute to the overall shaping of people’s perspectives and actions regarding feelings and health associated with eating. Bronfenbrenner’s ecological systems theory explains development within a series of layered, nested, interacting systems, including food choices [24]. According to health psychology research, dietary choices are impacted by cultural and societal norms in addition to individual characteristics [25]. This highlights the complex interaction of aspects that determine emotions associated with eating.

According to the Theory of Planned Behavior, attitudes, subjective norms, and perceived behavioral control all play a role in shaping people’s intentions and actions [25,26]. According to this notion, individuals’ subjective views of food-related emotions are influenced by their ideas and experiences. The theory’s focus on the subjective aspect of attitudes and perceptions in determining behaviors aligns with the diversity of cultural origins, personal preferences, and past experiences with food.

According to Erikson’s theory of psychosocial development [27], different age brackets come with distinctive psychological responsibilities. People in Erikson’s “Intimacy vs. Isolation” stage, which encompasses the years between 18 and 30, are navigating the process of developing close relationships. The connection between feelings about food and health may change due to new factors introduced throughout this developmental period, such as interpersonal interactions and personal lifestyle choices. The relationship’s dynamics could change when people encounter new psychosocial issues after this age range.

There are a lot of elements to think about when trying to figure out why people have specific feelings toward food and its relationship to health. Just like assembling a jigsaw, no one piece clarifies everything. Multiple factors contribute to the intricacy, including our cultural heritage, the individuals we associate with, and our distinct life experiences. This study may have resulted in a different picture had we focused on another age bracket. Nevertheless, there are many phases that everyone experiences. Relationships, for example, are typically worked out by people in this age bracket. Life presents new obstacles after this age bracket. People at different points may associate different feelings with certain foods.

A good analogy would be solving a puzzle where the components constantly shift. There is no universally applicable solution; as the study’s findings demonstrate, several circumstances impact people’s perceptions of the impact of their dietary choices on their well-being. Comparable to deciphering a multi-colored painting, one must examine each brushstroke to grasp the overall subject.

### 4.1. Implications and Recommendations

Various areas can benefit from the study’s insights. In order to promote healthier eating habits, health promotion programs could use the knowledge of the connection between food choices and well-being to their advantage. Based on these findings, public health initiatives could highlight the importance of food in relation to emotions and relationships. These findings can be incorporated into wellness and nutrition programs in educational institutions, raising knowledge about the psychological and emotional components of food. A more comprehensive strategy for the welfare of students may benefit from this. Family counselors can also use the study’s findings to help families work through their food-related problems. It is possible to improve family well-being through personalized interventions if we can identify how food choices affect relationships within families.

Several suggestions for future studies were derived from the study. Researchers should broaden the age range of participants in future studies beyond the 18–30 age bracket in order to better understand age-related differences. To further comprehend the dynamic nature of the connection between dietary preferences and health, longitudinal studies should be undertaken. Future studies should take cultural, social, and geographical differences into account using a more diverse sample to increase generalizability. Research findings could be made more robust by combining self-reported data with objective measurements like dietary evaluations or biomarkers. Finally, the study’s findings could inform intervention studies that seek to enhance people’s health by encouraging them to adopt better eating habits.

### 4.2. Limitations

It is important to take note of the limitations of this study. Firstly, the results may not be applicable to a wider age range because the participants had to be between the ages of 18 and 30. Secondly, there is always the chance of memory loss or social desirability bias when a study depends on self-reported data. The study’s validity could be enhanced by using a variety of data collection methods or by including objective measurements. The research also uses a cross-sectional approach, which provides a snapshot instead of a longitudinal perspective, making it harder to demonstrate causal correlations. Additionally, we acknowledge that online surveys may bring biases compared to face-to-face interviews, including self-selection and limited interaction with qualified staff. It is important to exercise caution when extrapolating the study’s results to communities with more diversity, as cultural and geographical factors may have impacted the results.

## 5. Conclusions

The CCCI was used to examine the complex relationship between food and well-being, dividing participants into low, medium, and high CCCI groups. Despite gender, education, and family income inequalities, chi-square testing showed no statistically significant associations. For the 18–30 age range analyzed, the Culinary Comfort and Connection Index appears independent of demographic characteristics. These findings highlight the complexities of the relationship between culinary tastes and general well-being, highlighting the importance of individual experiences and environmental circumstances. The study adds to our understanding of the nuances of food-related emotions and their impact on human well-being.

## Figures and Tables

**Table 1 nutrients-16-01865-t001:** Culinary Comfort and Connection Index (CCCI) and gender.

CCCI Category	Male	Female	Total
Low	15.3%	24.3%	39.6%
Medium	18.9%	24.3%	43.2%
High	3.6%	13.5%	17.1%
Total	37.8%	69.0%	100%

**Table 2 nutrients-16-01865-t002:** Culinary Comfort and Connection Index (CCCI) and educational level.

CCCI Category	Undergraduates	Postgraduates	Total
Low	27.0%	12.6%	39.6%
Medium	30.6%	12.6%	43.2%
High	12.6%	4.5%	17.1%
Total	70.3%	29.7%	100%

**Table 3 nutrients-16-01865-t003:** Culinary Comfort and Connection Index (CCCI) and family income.

CCCI Category	Low Income	Middle Income	High Income	Total
Low	16.2%	12.6%	10.8%	39.6%
Medium	22.2%	11.7%	9.0%	43.2%
High	8.1%	5.4%	3.6%	17.1%
Total	46.8%	29.7%	23.4%	100%

## Data Availability

The data presented in this study are available on request from the corresponding author due to ethical reasons.

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
