# Peer review of "Culinary Comfort: Understanding the Connection between Food and Well-Being"

_nutrients, 2024, doi:10.3390/nu16121865_

Round 1

Reviewer 1 Report

Comments and Suggestions for Authors

This an interesing reerch article with adequate novelty in a novel, very interesting topic. However, some points should be addressed.

- The authors should add subheadings (e.g. Background, Methods, Results, Discussion and Conclusions) based on the guidelines of the journal.

- The statements: "A significance level of 0.05 was chosen. We used Pearson Chi-Square and Likelihood Ratio tests." should be ommited from the abstract. It doesnot need these statements in the Abstract.

- As a last sentence in the Abstract, the authors should include a statement about what future studies could be done in the future based on their results.

- In the 2nd paragraph of introduction, the authors report that "...having family meals together can help...." To this point th authos should a short statement about well-recognized diettary pattern which include lifestyle such as Mediterranean Diet/Lifestyle or others, including some relevant references.

- At the end of the Introduction section, the authos should report briefly the main aim of their study.

- In the section 1.2, in the 2nd paragraph, please include another one relevant reference separately and after reference 19.

- In the section 1.2, in the 3rd paragraph, please include

In the section 2.1, in all the below paragraphs, the authors should include relevant references into the text.

- Do the authors used any inclusion and exclusion criteria? This should  strongly be recommended to be added in the Methods section.

- In the section 2.2, the authors should report the statistical analysis used.

- From my point of view, in Tables 1, 2 and 3, the symbol of percentage %, it could be better included into the cells of the table near the numbers.

- The Likelihood Ratio tests could be included altogether in an additiona Table.

- I do not see any multivariate logistic analysis into the results section. Is it feasible ans scientifically sound to be performed a such analysis, describing the results and including another table concerning them?

- The Discussion section should be a bit enriched in each paragraph, including some aditional relevant reference.

- In the section 4.2, the authors should include as limitation that this is an online-based survey. So, there is a higher increased for potential bias compared to a face-to-face interview collection of the data of the study between the participants and a qualified personnel. 

- At the end of the conclusions, the authors should proposed what future studies could be performed based on their experience by the present study.

Comments on the Quality of English Language

Moderate editing of English language required

Author Response

- This an interesing reerch article with adequate novelty in a novel, very interesting topic. However, some points should be addressed.

> Thank you for this positive feedback.

- The authors should add subheadings (e.g. Background, Methods, Results, Discussion and Conclusions) based on the guidelines of the journal.

> The subheadings have already been according to the guidelines of the journal. Your recommendations are not according to the guidelines. For the abstract, the guidelines state the following: “We strongly encourage authors to use the following style of structured abstracts, but without headings”.

- The statements: "A significance level of 0.05 was chosen. We used Pearson Chi-Square and Likelihood Ratio tests." should be ommited from the abstract. It doesnot need these statements in the Abstract.

> Deleted from the abstract.

- As a last sentence in the Abstract, the authors should include a statement about what future studies could be done in the future based on their results.

> Statement has been added.

- In the 2nd paragraph of introduction, the authors report that "...having family meals together can help...." To this point th authos should a short statement about well-recognized diettary pattern which include lifestyle such as Mediterranean Diet/Lifestyle or others, including some relevant references.

> We have added a statement on Mediterranean diet including a reference.

- At the end of the Introduction section, the authos should report briefly the main aim of their study.

> We have added the objective statement.

- In the section 1.2, in the 2nd paragraph, please include another one relevant reference separately and after reference 19.

> Further references have been added.

- In the section 2.1, in all the below paragraphs, the authors should include relevant references into the text.

> Further references have been added.

- Do the authors used any inclusion and exclusion criteria? This should  strongly be recommended to be added in the Methods section.

> Inclusion and exclusion criteria added.

- In the section 2.2, the authors should report the statistical analysis used.

> All statistical analyses are described in section 2.2.

- From my point of view, in Tables 1, 2 and 3, the symbol of percentage %, it could be better included into the cells of the table near the numbers.

> We have added the percentage symbols directly in the Tables.

- The Likelihood Ratio tests could be included altogether in an additiona Table.

> Because the p-values from the Likelihood Ratio tests were so close and because they did not add anything meaningful, we decided to skip them. The primary focus instead was on the Pearson Chi-Square tests, which kept things simple and easy to understand. It was determined that there was no need to include the tests in a separate table.

- I do not see any multivariate logistic analysis into the results section. Is it feasible ans scientifically sound to be performed a such analysis, describing the results and including another table concerning them?

> Instead of a multivariate logistic analysis, our study examined individual correlations between demographic characteristics (gender, education, and family income) and Culinary Comfort and Connection Index (CCCI) categories. This method enables us to examine each demographic factor’s impact clearly and interpretably on participants' food-related feelings and well-being. Due to the intricacy of human dietary behavior and well-being, evaluating these elements independently revealed their unique contributions to the outcomes of interest. Our sample size and study design also encouraged univariate analysis rather than multivariate models with many predictor factors. We presented the results of individual studies to provide a thorough understanding of demographic characteristics and CCCI categories while making our findings accessible and interpretable for scholars and practitioners.

- In the section 4.2, the authors should include as limitation that this is an online-based survey. So, there is a higher increased for potential bias compared to a face-to-face interview collection of the data of the study between the participants and a qualified personnel. 

> We have added elaboration on the limitations of the study.

- At the end of the conclusions, the authors should proposed what future studies could be performed based on their experience by the present study.

> Suggestions for future study are already given in the implications and recommendations.

Reviewer 2 Report

Comments and Suggestions for Authors

The manuscript seems to align well with Bronfenbrenner’s Societal-Ecological Model, with the Theory of Planned Behavior playing a somewhat tangential role and Erikson showing up right at the end. The result is a substantial bit of conceptual clutter. Friendly advice is to drop Erickson altogether and to provide a clearer conceptual bridge between Bronfenbrenner and TPB.

The dataset is not really described in any detail, so it is difficult to know whether it is reasonably representative of the underlying population. For other researchers to be able to replicate the study's findings, substantially more information is needed about data collection and sample structure.

The manuscript nicely addresses implications of the findings.

Comments on the Quality of English Language

Only minor editing is needed.

Author Response

The manuscript seems to align well with Bronfenbrenner’s Societal-Ecological Model, with the Theory of Planned Behavior playing a somewhat tangential role and Erikson showing up right at the end. The result is a substantial bit of conceptual clutter. Friendly advice is to drop Erickson altogether and to provide a clearer conceptual bridge between Bronfenbrenner and TPB.

> Erickson is not used in the study.

The dataset is not really described in any detail, so it is difficult to know whether it is reasonably representative of the underlying population. For other researchers to be able to replicate the study's findings, substantially more information is needed about data collection and sample structure.

> Our methodology section provides all information to understand the general approach and procedures used in data collection and sample selection. We clearly outline the use of an online-based survey and describe the demographic factors considered, including gender, education, and household income.

The manuscript nicely addresses implications of the findings.

> Thank you for this positive feedback.

Round 2

Reviewer 1 Report

Comments and Suggestions for Authors

The authors have significantly improved their manuscript and their paper meets the criteria for publication.

It would be a pleasure for me to read and  propose my suggestions for the impovement of this manuscript. This article is well-written and well-organized and it concerns an interesting topic with high quality and novelty.

Reviewer 2 Report

Comments and Suggestions for Authors

The revised manuscript represents an improvement.

Comments on the Quality of English Language

The writing is clear enough and understandable, so just routine copyediting should be needed.